# Awareness, Knowledge, Perceptions, and Attitudes towards Familial and Inherited Cancer

**DOI:** 10.3390/medicina58101400

**Published:** 2022-10-06

**Authors:** Lolowah Alghuson, Nora I. Alturki, Allulu Saad Alsulayhim, Luluh Y. Alsughayer, Khalid M. Akkour

**Affiliations:** 1Obstetrics and Gynecology Department, College of Medicine, King Saud University, King Saud University Medical City, Riyadh 11461, Saudi Arabia; 2Internal Medicine Department, College of Medicine, King Saud University, King Saud University Medical City, Riyadh 11461, Saudi Arabia; 3Otorhinolaryngology Department, College of Medicine, King Saud University, King Saud University Medical City, Riyadh 11461, Saudi Arabia

**Keywords:** familial and inherited cancer, genetic testing, awareness

## Abstract

*Background and Objectives*: In 2020, the World Health Organization (WHO) reported 9.9 million deaths from cancer, with a mortality rate of 10.65%. Early detection of cancer can decrease mortality and increase the chance of cure. In Saudi Arabia, multiple studies were performed for awareness and attitudes toward cancer, but few studies evaluated the awareness of familial and inherited cancers. *Materials and Methods*: This is a cross-sectional observational survey of the awareness, knowledge, and attitudes of Saudi women toward familial and inherited cancers. The estimated sample size was 385. Questionnaires were distributed through social media platforms from 1 January 2021 to 22 January 2021. *Results*: Of the 385 participants, the majority have a bachelor’s degree. More than half (68.9%) know that family history is related to cancer, and approximately 57.2% are aware of genetic testing. The most common indication of genetic testing is premarital testing (18.5%). An inverse relationship is noted between the awareness of familial and inherited cancers and age (*p* = 0.003, CI = 0.723–0.938). However, awareness of inherited and familial cancer is positively associated with awareness of the association of genetic mutation to cancer (*p* = 0.013, CI = 1.080–1.921) and knowledge about genetic testing (*p* > 0.000, CI = 2.487–8.426). *Conclusions*: Our results reveal that Saudi women, especially older adults, have suboptimal knowledge about inherited and familial cancers, and poor attitudes toward genetic screening. We recommend increasing public awareness regarding risk factors and screening for inherited and familial cancers.

## 1. Introduction

Cancer is a common cause of morbidity and mortality. In 2020, breast, lung, colorectal, prostate, stomach, and liver cancers were the most common cancers worldwide [1]. The World Health Organization (WHO) reports 19.29 million new cancer cases every year [1]. Malignancy is the second most common cause of death, accounting for approximately 20% of all deaths in 2017 [2]. In 2020, the WHO reported 9.9 million cancer deaths (mortality rate: 10.65%) [1]. Cancer also has a substantial economic burden; it was estimated that the national cancer care in 2020 will cost USD 157.77 billion [3].

In Saudi Arabia, the number of newly diagnosed cancer cases in 2020 was 27,885 (14,253 were males and 13,632 were females). The most common cancers in males were colorectal cancer, non-Hodgkin lymphoma, leukemia, and thyroid and lung cancers. In females, the most common types of cancer were breast, thyroid, colorectal, and uterine cancers, and leukemia. The number of deaths due to cancer was 13,069 (mortality rate: 5.41%) [4].

Given the huge impact of malignancy, early cancer detection and treatment initiation can decrease mortality and increase the chance of cure [5]. The most common method to detect cancer early is screening, such as a mammogram for breast cancer, colonoscopy for colorectal cancer, or Pap smear for cervical cancer [6,7,8]. Familial and inherited cancer can present at an early age; thus, screening for these cancers should be performed earlier than in the general population [9]. Early screening decreases cancer mortality and morbidity [10].

Many studies addressed the Saudi populations’ awareness and attitude toward cancer. Awareness of breast cancer among Saudi women was found to be poor [11,12]. Misinformation about mammographic screening was found to significantly reduce its use by 56% (OR = 0.44; 95% CI = 0.22–0.88) [12]. In addition, Saudis had poor-to-moderate awareness of cervical cancer [13,14]. While a study in Qassim shows negative attitudes toward Pap screening and human papillomavirus (HPV) vaccination [13], another study shows a preference for adding HPV testing to premarital screening [14]. A survey about colorectal cancer awareness in Riyadh reveals some misconceptions regarding universally accepted screening protocols, symptoms, and a general understanding of colorectal cancer in Saudi Arabia [15].

There are many etiologies of cancer; one of them is inherited/familial, such as Lynch syndrome [16] and BRCA1 and 2 gene mutations [17]. About 30% of patients with colorectal cancer have a family history of colorectal cancer [18], and a positive family history is associated with a worse prognosis in patients with head and neck cancer than in those without a family history [19].

Despite the availability and tangibility of information in the Internet era, worldwide communities had poor–moderate knowledge about inheritance features and factors of breast cancer [20,21] and prostate cancer [22]. It was found that awareness of inherited prostatic cancer was associated with elder age and having a family member with cancer. However, no association was found in education level [22]. Meanwhile, there was a significant association between age and level of education in the knowledge of inherited breast cancer in Saudi women [23]. A systematic review shows that ethnic minorities have lower knowledge of inherited cancer and genetic testing [24]. Roberts et al. show that awareness of genetic testing is significantly associated with non-Hispanic, middle age, and higher education levels [25] This study assessed the awareness and attitude of Saudis toward familial and inherited cancer, as well as the associated factors.

## 2. Materials and Methods

This was a cross-sectional observational survey of the awareness, knowledge, and perception of Saudi women toward familial and inherited cancers. On the basis of the Saudi female census of 2018, which included 10,192,732 females, the sample size was estimated. With a confidence level of 95% and a confidence interval of 5%, the approximate sample size was 385.

There were two major sections of close-ended elements in the questionnaire. The first section included sociodemographic data, and the second evaluated the respondents’ awareness of familial and inherited cancer, adapted from a study in Jordan [26,27]. A pilot with 50 participants was conducted to modify the survey according to the study objectives and review for consistency. The electronic questionnaire was shared on social media platforms to recruit participants through convenient sampling. The inclusion criteria involved Saudi women > 17 years old. Data collection started on 1 January 2021, and was concluded at the completion of the sample size on 22 January 2021. 

Data were then exported for analysis using SPSS for Windows, version 21.0 (SPSS Inc., Chicago, IL, USA). Categorical variables are expressed as percentages. The odds ratio was used to study association between different variables. Univariate and multivariate logistic regression were used to assess the predictive factors of the awareness of familial and inherited cancer among Saudis, with *p* < 0.05 considered significant. 

The study was approved by the King Saud University Institutional Review Board (E-20-5397). The participants completed the online surveys anonymously and voluntarily. Furthermore, each participant had a unique IP address to prevent duplication, and the participants were not offered rewards or incentives for their participation.

## 3. Results

A total of 385 participants completed the questionnaire. Most participants are 18–24 years (35%), have a bachelor’s degree (65%), are single (55.9%), are unemployed (77.2%), and have no health insurance (68.4%). Only 3.5% of them are smokers, and 32.7% have children (Table 1).

Table 2 describes the awareness of these participants toward familial and inherited cancer, manifested by awareness of the association of cancer with consanguinity (21.8%), family history (68.9%), and genetic mutations (57.5%), along with the association of consanguinity with congenital malformations (79.2%). More than half of these participants know about genetic testing (57.2%), but only a quarter (25.32%) have undergone some sort of genetic testing. The most common indication of genetic testing is premarital testing (18.5%), as requested by the court (15.7%). Most of those who partake in the test (11.4%) are neither briefed about or consented to it before performing the test, but are generally satisfied (16.8%) with the level of privacy offered. Furthermore, 57% are willing to be screened for cancer through genetic testing, but only 21.8% know about the role of the genetic counselor.

There is a huge variation in the awareness of the participants toward familial and inherited cancer in different cities in Saudi Arabia (Figure 1).

Awareness of familial and inherited cancer is associated with only a few factors, including younger age (OR = 0.863, *p* = 0.017, CI = 0.765–0.973), awareness of the association of genetic mutation with cancer (OR = 1.542, *p* = 0. 002, CI = 1.169–2.034), and the knowledge about genetic testing (OR = 4.291, *p* = 0. 000, CI = 2.385–7.718) (Table 3). However, some of the factors have a wide CI, which can indicate less precision of our sample. In multivariate analysis, younger age (OR = 0.823, *p* = 0.003, CI = 0.723–0.938), awareness about the association of genetic mutation with cancer (OR=1. 441, *p* = 0.013, CI = 1.080–1.921), and knowledge about genetic testing (OR = 4. 578, *p* > 0.000, CI = 2.487–8.426) are found to be significant for awareness of familial and inherited cancer (Table 4).

## 4. Discussion

This study addressed the awareness and attitude of Saudi women toward inherited cancer and assessed the associated factors. Although most participants are educated (with a bachelor’s degree), most have suboptimal knowledge about inherited and familial cancers. Moreover, an inverse relationship is observed between awareness of familial and inherited cancers and age; older participants are significantly less aware (OR = 0. 823, *p* = 0.003, CI = 0.723–0.938).

Saudi Arabia has a high rate of consanguinity (56%), with some regional variations, and the highest prevalence being in rural areas [28]. Consanguineous marriage is proven to be associated with congenital anomalies [29,30] as well as specific cancers, such as leukemia, lymphoma, colorectal, and prostate cancers [31]. In this study, 53.4% of the study participants thought that cancer has no association with consanguinity, but they are more aware of the association of consanguinity with congenital anomalies (79.2%). This highlights the importance of educating the population about the risks of consanguinity and its effect on inherited diseases.

Most participants are aware of the association of family history (68.9%) and genetic mutations (57.5%) with cancer. Although more than half of the respondents (57.2%) know that there are genetic tests to detect familial and inherited cancers, the majority (74.2%) have never undergone genetic testing. Costs and lack of insurance are reported as major factors that discourage genetic testing [32,33,34]. Most of our participants are unemployed (77.2%), and have no health insurance (68.4%).

Of the 105 participants who underwent genetic testing, 59% participated for premarital testing (which was under the Healthy Marriage Program), and only 7.6% through a family physician. The low tendency of participants toward genetic testing could be linked to the fact that most participants (68.4%) do not have insurance to cover it. Furthermore, only 21.8% understand the role of the genetic counselor. The Healthy Marriage Program is a national premarital screening program for some of the highly prevalent inherited diseases in Saudi Arabia, such as sickle cell disease and thalassemia [35]. Only 73 participants (18.5%) have undergone premarital testing. However, this might be because the program was initiated and mandated in 2004.

Family physicians and general practitioners play a vital role in referring at-risk patients to genetic counselors or clinical geneticists. However, one study shows that general practitioners have limited knowledge about hereditary cancers, testing services, and genetic counseling [36], which may delay the referral of high-risk patients and their families to specialized care [36].

The attitudes of our participants toward genetic screening are not reassuring: 43% are unwilling to undergo cancer screening through genetic testing. Further efforts are required to ensure the proper counseling, consenting, and privacy of those undergoing genetic testing.

This study has a few limitations. The study design is a cross-sectional study, which may be prone to incidence–prevalence bias. The sample size only includes women, who happen to be mostly bachelor’s degree graduates. Therefore, the results of this study cannot be generalized to the general Saudi population. Also, there is an inflation of the CI due to the use of 95% CI, therefore, the problem of the monotonic likelihood of the univariate model may be noted in this study [37]. We recommend future cohort studies addressing this topic to include a more representative sample with a larger sample size. 

## 5. Conclusions

In conclusion, our findings indicate suboptimal knowledge among Saudi women, particularly older women, about inherited and familial cancers, and poor attitudes toward genetic screening. We recommend increasing public awareness regarding risk factors and screening for inherited and familial cancers.

## Figures and Tables

**Figure 1 medicina-58-01400-f001:**
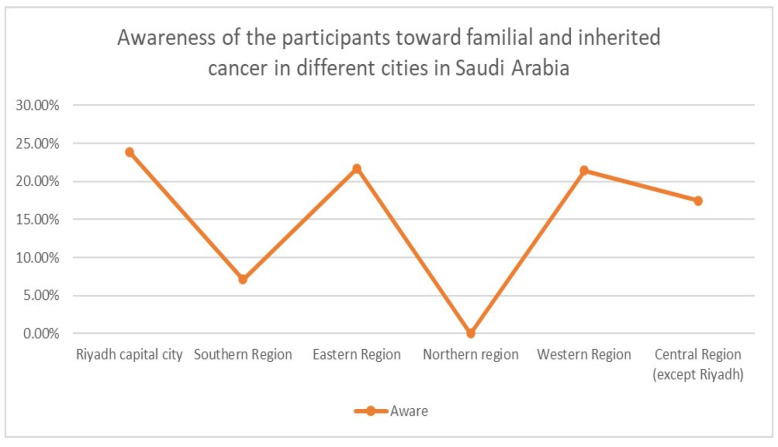
Awareness of the participants toward familial and inherited cancer in different regions in Saudi Arabia. *X* axis: main regions in Saudi Arabia. *Y* axis: percentage of the awareness toward familial and inherited cancer.

**Table 1 medicina-58-01400-t001:** Sociodemographic data.

Variables	Frequency (%)
**Age (in years)**	
<18	51 (12.9)
18–24	141 (35.7)
25–34	113 (28.6)
35–44	58 (14.7)
45–54	24 (6.1)
55–64	8 (2)
**Education**	
Elementary school	3 (0.8)
Middle school	19 (4.8)
High school	91 (23)
Bachelor’s degree	258 (65.3)
Post–graduate degree	24 (6.1)
**Smoking cigarettes or shisha**	
No	380 (96.2)
Yes	15 (3.8)
**Social status**	
Single	221 (55.9)
Married	158 (40)
Divorced	10 (2.5)
Widow/widower	6 (1.5)
**Are you employed?**	
No	305 (77.2)
Yes	90 (22.8)
**Family income**	
<5000 SAR	60 (15.2)
5000–10,000 SAR	117 (29.6)
10,000–20,000 SAR	48 (12.2)
200,000–30,000 SAR	134 (33.9)
>30,000 SAR	36 (9.1)
**Do you have Kids?**	
No	266 (67.3)
Yes	129 (32.7)
**Do you have insurance?**	
No	270 (68.4)
Yes	125 (31.6)

**Table 2 medicina-58-01400-t002:** Awareness, perceptions, and attitudes towards familial and inherited cancer (n = 395).

Survey Item	Frequency (%)
**Q1 (Do you think consanguineous marriage is relevant to cancer?)**	
No	211 (53.4)
I do not know	98 (24.8)
Yes	86 (21.8)
**Q2 (Do you think family history is relevant to cancer?)**	
No	67 (17)
I do not know	56 (14.2)
Yes	272 (68.9)
**Q3 (Do you think genetic mutations are relevant to cancer?)**	
No	56 (14.2)
I do not know	112 (28.4)
Yes	227 (57.5)
**Q4 (What do you think of the phrase: “parents’ consanguinity is relevant to congenital malformations in offspring")**	
I disagree	39 (9.9)
I highly disagree	6 (1.5)
I agree	192 (48.6)
I highly agree	121 (30.6)
I do not know	37 (9.4)
**Q5 (Have you ever heard or read about genetic tests?)**	
No	169 (42.8)
Yes	226 (57.2)
**Q6 (What type of genetic testing did you have?)**	
Diagnostic testing	15 (3.8)
Carrier testing	4 (1)
Predictive and pre-symptomatic testing	1 (0.3)
Premarital testing	73 (18.5)
Do not know	12 (3)
I did not do test	295 (74.7)
**Q7 (Who requested this genetic test?)**	
Nobody	10 (2.5)
General/family physician	8 (2)
Specialist physician	6 (1.5)
Clinical geneticist	3 (0.8)
Genetic counselor	1 (0.3)
Court (marital request)	62 (15.7)
Other	4 (1)
Do not know	11 (2.8)
I did not do test	295 (74.7)
**Q8 (Did the requester use a consent form or brief you verbally about genetic testing, before ordering the test?)**	
No	45 (11.4)
Yes, briefed me verbally	24 (6.1)
Yes, gave me a consent form	10 (2.5)
Yes, both verbally and consent form	5 (1.3)
Do not know	16 (4.1)
I did not do the test	295 (74.7)
**Q9 (How would you rate your level of satisfaction with the level of privacy offered to you, during the genetic testing?)**	
Very satisfied	33 (8.4)
Satisfied	33 (8.4)
Dissatisfied	3 (0.8)
Very dissatisfied	0 (0)
Do not know	31 (7.8)
I did not do the test	295 (74.7)
**Q10 (How likely is it that you choose to undergo a genetic test to know your risk of developing cancer?)**	
Unlikely	123 (31.1)
Very unlikely	47 (11.9)
Likely	161 (40.8)
Very likely	64 (16.2)
**Q11 (Have you ever heard or read about genetic counselors?)**	
No	309 (78.2)
Yes	86 (21.8)

**Table 3 medicina-58-01400-t003:** Univariate analysis for the assessment of the factors influencing awareness of family and inherited cancer.

	B	S.E.	*p*-Value	OR	95% CI	
					Lower	Upper
City	−0.087	0.055	0.114	0.917	0.823	1.021
Age	−0.147	0.061	0.017	0.863	0.765	0.973
Educational level	0.060	0.138	0.663	10.062	0.810	1.392
Smoking	0.119	0.537	0.824	10.127	0.393	3.229
Social status	−0.045	0.178	0.802	0.956	0.675	1.355
Employment	0.268	0.246	0.275	10.308	0.807	2.119
Family income	−0.013	0.078	0.864	0.987	0.846	1.151
Having children	0.103	0.218	0.635	10.109	0.724	1.698
Having health insurance	0.363	0.222	0.103	10.437	0.930	2.221
Do you think consanguineous marriage is relevant to cancer?	0.217	0.128	0.089	10.243	0.967	1.597
Do you think family history is relevant to cancer?	0.256	0.132	0.052	10.292	0.997	1.674
Do you think genetic mutations are relevant to cancer?	0.433	0.141	0.002	10.542	10.169	2.034
What do you think of the phrase: “parents’ consanguinity is relevant to congenital malformations in offspring”	0.138	0.101	0.174	10.148	0.941	1.400
Did you ever undergo a genetic test?	0.151	0.121	0.214	10.163	0.917	1.475
How likely is it that you choose to undergo a genetic test to know your risk of developing cancer?	0.087	0.093	0.351	10.091	0.909	1.310
Have you ever heard or read about genetic counselors?	10.456	0.300	0.000	40.291	20.385	7.718

**Table 4 medicina-58-01400-t004:** Multivariate analysis of significant variables influencing awareness of family and inherited cancer (*p* < 0.05).

	B	S.E.	*p*-Value	OR	95% CI	
					Lower	Upper
Age	−0.194	0.067	0.003	0.823	0.723	0.938
Do you think genetic mutations are relevant to cancer?	0.365	0.147	0.013	1.441	1.080	1.921
Have you ever heard or read about genetic counselors?	1.521	0.311	0.000	4.578	2.487	8.426

## Data Availability

The data used to support the findings of the study can be available at request to the corresponding author.

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
