# Peer review of "Awareness, Knowledge, Perceptions, and Attitudes towards Familial and Inherited Cancer"

_medicina, 2022, doi:10.3390/medicina58101400_

Round 1
Reviewer 1 Report
The article is well presented with moderate editing needed
The methodology should be a bit clearer.
although more results would be nice , I think the study is important. Thus I recommend it’s publication.
Author Response
Thank you for your review and comments.
The manuscript was reviewed and paraphrased through native-English experienced editors at Screbendi.com. The methodology is now revised to briefly summarizes study methodology and statistical analysis. The results section reflects all results pertinent to study findings.
Please let us know if you have any further comments, we’ll be happy to adjust accordingly.
Reviewer 2 Report
The paper provides an insight into the awareness of familial and inherited cancers in Saudi Arabia. It is of interest due to the lack of epidemiological information of this WHO region.
Introduction is fine and provides background related with the proposed study. Methods of analysis are simple, but these are also well described. In the results section, however, the authors provide “associations” or “relationships” based on confidence intervals and p-values, see “An inverse relationship was noted between the awareness of familial and inherited 26 cancers and age (p = 0.003, CI = 0.723–0.938). However, awareness of inherited and familial cancer 27 was positively associated with awareness of the association of genetic mutation to cancer (p = 0.013, 28 CI = 1.080–1.921) and knowledge about genetic testing (p > 0.000, CI = 2.487–8.426).”
Please provide the measure of association: Odds ratios in Abstract and in results section (Exp(B) must be Odds ratio, also in Table 3).
There is a limitation on the study related with “Of the 385 participants, the majority had a bachelor’s degree”. Based on descriptive presented in Table 1, the conclusions are driven mainly from the subsample of 65.3% (N=258) women with bachelor degree. Authors must note in the discussion that this could be a limitation of the study towards a bias. Authors must note that in future studies, there must be compared results according to education degree.
Finally, the authors must write a legend for Table 1 (such as the legend for Figure 1), and improve the legend of Table 2. Table 3’s legend might be re-writted as “Table 3: Univariate analysis for the assessment of the factors influencing awareness of Family & Inherited Cancer”.
Reviewer 3 Report
This is a cross-sectional observational study discussing about the awareness and attitude of Saudis toward familial and inherited cancer as well as the associated factors. I think the topic is interesting and contributive to the public health under an empirical approach quite valuable for public social policy professionals.
First, please remove the “0. How to Use This Template” from manuscript.
Major concerns:
1. Introduction: The author should provide some review or meta-analysis (if any) to present perspective of familial and inherited cancer as well as the associated factors.
2. Introduction: Does any socioeconomic inequalities known for the familial and inherited cancer as well as the associated factors (i.e., Age)?
3. Method: The author should provide correlation analysis of associated factors and outcome in this study?
4. Method: It seems poor description of statistical analysis? Mean or median presentation for continuous variables? The author should clarify this concern.
5. Method: How is the reliability of used questionnaire?
6. Results: Why the confidence intervals of ORs are wider? The author should contribute more information to readers. I mean inflation of ORs in Table 3.( PMCID: PMC8274998)
7. Results: There still some confounding factor may disturb the results. For example, Age, Charlson Comorbidity Index, sex, and so on. Any adjusted model for logistic regression model?
8. Discussion: Social determinants of health inequalities associated with disease were well known in previous studies. The author should clarify this concern.
9. Discussion: To our knowledge, we examined that the use of this design (i.e., cross-sectional study) is subject to Incidence-Prevalence bias, also known as Neyman bias, and how that might influence their findings.
Round 2
Reviewer 3 Report
Thanks for your efforts on revision.
Minor for adding a limitation about inflation of 95% CI of odds ratio.
The problem of the monotonic likelihood of the univariate model may be noted in this study [ref].
ref: Tzeng, I.-S. Dealing with the Problem of Monotone Likelihood in the Inflation of Estimated Effects in Clinical Studies. Comment on Hasegawa et al. Impact of Blood Type O on Mortality of Sepsis Patients: A Multicenter Retrospective Observational Study. Diagnostics 2020, 10, 826. Diagnostics 2022, 12, 2295. https://doi.org/10.3390/diagnostics12102295
Author Response
Thanks for your efforts on revision.
Minor for adding a limitation about inflation of 95% CI of odds ratio.
The problem of the monotonic likelihood of the univariate model may be noted in this study [ref].
ref: Tzeng, I.-S. Dealing with the Problem of Monotone Likelihood in the Inflation of Estimated Effects in Clinical Studies. Comment on Hasegawa et al. Impact of Blood Type O on Mortality of Sepsis Patients: A Multicenter Retrospective Observational Study. Diagnostics 2020, 10, 826. Diagnostics 2022, 12, 2295. https://doi.org/10.3390/diagnostics12102295
Thank you for your comment. A new limitation has been added according to your comment. We are not quite sure we understood the concept very well, so if you have any further comments regarding the sentence added we would be more than happy to hear it.